# The Importance of Mycorrhizal Fungi in the Development and Secondary Metabolite Production of *Echinacea purpurea* and Relatives (Asteraceae): Current Research Status and Perspectives

Martin Iakab [1,2], Erzsébet Domokos [2], Klára Benedek [2], Katalin Molnár [2], Endre Kentelky [2], Erzsebet Buta [3] and Francisc Vasile Dulf [1,*]

[1] Department of Engineering and Environmental Protection, University of Agricultural Sciences and Veterinary Medicine Cluj-Napoca, 3-5 Mănăștur Street, 400372 Cluj-Napoca, Romania
[2] Department of Horticulture, Sapientia Hungarian University of Transylvania, Sighișoarei 2, 540485 Târgu Mureș, Romania
[3] Department of Horticulture and Landscaping, University of Agricultural Sciences and Veterinary Medicine Cluj-Napoca, 3-5 Mănăștur Street, 400372 Cluj-Napoca, Romania
* Correspondence: francisc.dulf@usamvcluj.ro

**Abstract:** The cultivation of *Echinacea purpurea* for commerce and obtaining high-quality plant material on a large scale remain a challenge for growers. Another challenge for the following decades is to create sustainable agriculture that meets society's needs, has no environmental impact, and reduces the use of fertilizers and pesticides. The aims of this overview were: (1) to present the importance of the chemical compounds reported in *E. purpurea*; (1) to synthesize results about cultivation of the *E. purpurea* with arbuscular mycorrhizal fungi (AMF) and associated microorganisms; (2) to exemplify similar research with plants from the Asteraceae family, due to the limited number of published *Echinacea* studies; (3) to collect recent findings about how the inoculation with AMF affects gene expressions in the host plants; (4) to propose perspective research directions in the cultivation of *E. purpurea*, in order to increase biomass and economic importance of secondary metabolite production in plants. The AMF inocula used in the *Echinacea* experiments was mainly *Rhizophagus irregularis*. The studies found in the selected period (2012–2022), reported the effects of 21 AMFs used as single inocula or as a mixture on growth and secondary metabolites of 17 plant taxa from the Asteraceae family. Secondary metabolite production and growth of the economic plants were affected by mutualistic, symbiotic or parasitic microorganisms via upregulation of the genes involved in hormonal synthesis, glandular hair formation, and in the mevalonate (MVA), methyl erythritol phosphate (MEP) and phenylpropanoid pathways. However, these studies have mostly been carried out under controlled conditions, in greenhouses or in vitro in sterile environments. Since the effect of AMF depends on the variety of field conditions, more research on the application of different AMF (single and in various combinations with bacteria) to plants growing in the field would be necessary. For the identification of the most effective synergistic combinations of AMF and related bacterial populations, transcriptomic and metabolomic investigations might also be useful.

**Keywords:** alkamides; Asteraceae; caffeic acid derivatives; *Echinacea purpurea*; glycoproteins; mycorrhiza; polysaccharides; secondary metabolite production

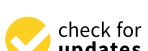



## 1. Introduction

The purple coneflower (*Echinacea purpurea* (L.) Moench) from the Asteraceae (Compositae) family is a perennial ornamental and medicinal plant native to North America, known primarily for its role in the immune system.

In recent decades, it has received increasing attention in pharmaceutical studies due to the diverse bioactive constituents it contains. Chemical compounds reported from

*Echinacea purpurea* include phenolic compounds (phenolic acids and flavonoids), alkamides, essential oil, polysaccharides, and glycoproteins [1–4]. Scientific research has suggested that alkamides, glycoproteins, polysaccharides, and caffeic acid derivatives (CADs) are responsible for *Echinacea*'s immunostimulatory activity [2–5].

*E. purpurea* is one of the most widely used herbs in traditional medicine for the treatment of respiratory diseases by stimulating the immune system [6–8]. Herbal end products are regulated by national pharmacopeial standards [9–11] and European Union regulations [12]. According to Munteanu et al. [13], *E. purpurea* is commonly used to prepare capsules, extracts and tinctures for herbal supplementation. The most common herbal parts in trade are leaves and above-ground parts, *E. purpurea* herba, but roots, *E. purpurea* radix, are also listed in the European Scientific Cooperative on Phytotherapy Monographs [12,14].

*E. purpurea* grows best in soils with a pH of 6 to 7. Drought cycles and plant stress are thought to raise levels of beneficial constituents [15]. In general, dry, low-nitrogen soils produce higher concentrations of essential oils. Wet, nitrogen-rich soils produce high levels of alkaloids [16]. Differences in soil type (sandy or loamy) and fertilization regime also have an impact on the presence and amount of phenolic compounds [17].

In its native habitat, *E. purpurea* is frost-resistant and is a hardy perennial, and it can tolerate temperatures of −25 °C to −40 °C if there is a snow layer. In Europe, *Echinacea* winters safely throughout southern and central Europe [15]. While *E. angustifolia* develops only basal leaves in the first year and develops flowers only in the second year, field-grown *E. purpurea* populations bloom in a proportion of 40–60% in the first year [13].

*E. purpurea* can be cultivated by sowing and transplanting. Despite the many advantages of transplanting over direct sowing, it is still neglected in practice because of its time-consuming requirements [13]. The low germination rate and the seed dormancy cause further difficulties in large-scale production [18,19]. Although micropropagation has great potential for producing plants with large amounts of plant material and enhanced secondary metabolites, acclimatization of plants is proving difficult [20,21]. Therefore, the cultivation of *E. purpurea* for economic purposes and obtaining high-quality plant material on a large scale remain a challenge for growers.

Another challenge for the following decades is to create sustainable agriculture that meets society's needs, has no environmental impact, and reduces the use of fertilizers and pesticides. Beneficial soil micro-organisms, which play a key role in maintaining long-term soil fertility and health, reducing chemicals in agriculture, providing plant nutrition and producing safe and quality crop products, can provide a solution to achieving these objectives [22].

Endophytes are microbial species that colonize plants without causing disease and are associated with almost all terrestrial plants. Of the endophytes, arbuscular mycorrhizal fungi (AMF) have the most extensive plant symbiosis, being able to colonize the roots of more than 80% of terrestrial plants through hyphal networks. AMF's hyphae penetrate the interior of the root cells, but they do not enter through the plasma membrane into the symplast. *Arum*-type mycorrhizae are present in the intercellular spaces as well, and form in the cells' terminal arbuscules. Several species also have vesicles in addition to arbuscules, which are older hyphal structures. *Paris*-type mycorrhizae present an intracellular growth, forming coils of hyphae and intercalary arbuscules. The host plant absorbs from the arbuscules the accumulated minerals [22,23]. Approximately 320 AMF species have been classified in the phylum Glomeromycota, but relatively few species were studied in their interactions with various host plants [22,24,25]. In this mutualistic–symbiotic relationship, the fungus supports the nutrients uptake (especially phosphorus) of the roots through the mycelia by providing a large surface area, and in return receives carbohydrates (photosynthesized sugars), vitamins and amino acids from the plant as organic nutrients [22,26–30].

Recent studies have shown that AMF and their associated bacterial communities (the mycorrhizospheric bacteria) are able to change the quantity and quality of secondary metabolic products in the host plant. In addition, these studies also described changes

in other properties of the plants, e.g., biomass increase, better nutrient uptake and water balance, increase in glandular hairs density, changes in the synthesis of plant hormones, and improved resistance to stress [22,30,31].

The aims of this overview were: (1) to present the importance of the chemical compounds reported in *E. purpurea*; (1) to synthesize results about cultivation of *E. purpurea* with AMF and associated microorganisms; (2) to exemplify similar research with plants from the Asteraceae family, due to the limited number of published *Echinacea* studies; (3) to collect recent findings about how the inoculation with AMF affects gene expressions in the host plants; (4) to propose perspective research directions in the cultivation of *E. purpurea*, in order to increase biomass and economic importance secondary metabolite production in plants.

The specialty literature (24 articles) included in this overview was collected from the Science Direct, Springer Link and Google Academic platforms (Figure 1). Because of the small number of literature studies in the case of the genera *Echinacea* and the species *Echinacea purpurea* cultivated with the mutualistic or symbiotic microorganisms (five articles), all times works were used. The terms of search were the following: *Echinacea*, *Echinacea purpurea*, mycorrhyza, AMF, growth, chemical compounds, bioactive compounds, bioactive principles, and secondary metabolites. The search was extended also on the species from the Asteraceae family. In this case, the literature of the years 2012–2022 (13 articles) were included. In order to collect the recent findings about how the inoculation with AMF affects gene expressions in different host plants, the literature of the past 10 years was consulted, and six articles were selected. The search terms were the following: mycorrhiza, AMF, gene expression, and mechanism of action.

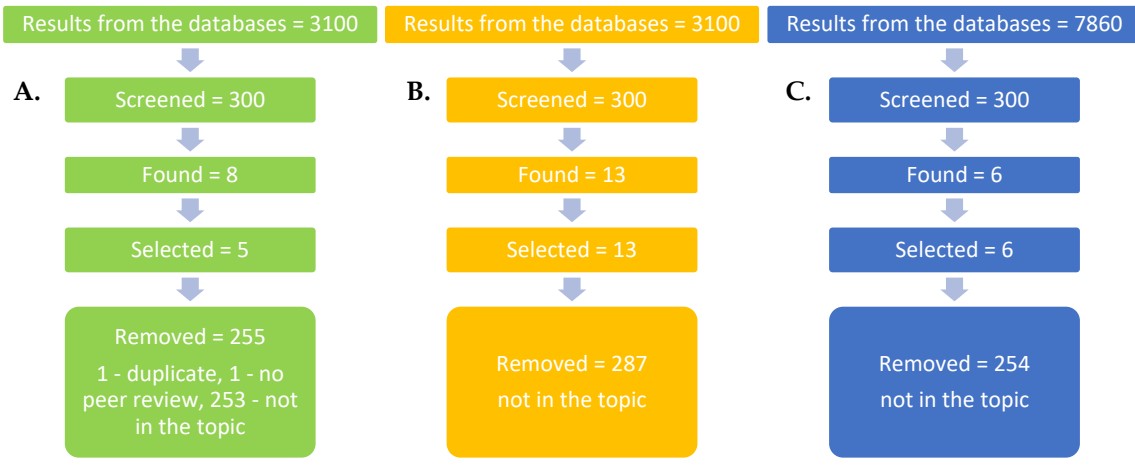

**Figure 1.** Literature research and article collection: (**A**) *Echinacea* cultivated with arbuscular mycorrhiza for enhancing growth and secondary metabolite production. (**B**) Plants from Asteraceae family cultivated with arbuscular mycorrhiza for enhancing growth and secondary metabolite production. (**C**) Arbuscular mycorrhiza effect on gene expressions in different host plants.

## 2. Importance of Chemical Compounds Reported in *Echinacea purpurea*

Numerous phytochemicals have been detected in *E. purpurea*. The most relevant of these compounds are alkamides, caffeic acid derivatives (mainly chicoric acid, chlorogenic acid, caftaric acid, cynarin, and echinacoside) [32–34], essential oil (predominantly borneol, carvomenthene, β -caryophyllene, myrcene, limonene, germacrene D, α- and β-pinen) [1,35–38], glycoproteins and polysaccharides [1,39–41].

### 2.1. Alkamides

The alkamides found in *E. purpurea* [42,43] are presented in Figure 2. Fatty acid amides possess numerous biological activities [44], and are involved in signaling pathways relevant to inflammation, pain, cancer and cardiovascular disease. Furthermore, Cruz et al. [45]

suggested that alkamides may play a role in disrupting the fungal cell wall/membrane complex. A study conducted by Chicca et al. [46] reported that N-alkylamides exert modulatory effects on the endocannabinoid system by simultaneously targeting endocannabinoid transport and degradation as well as the human cannabinoid receptor type-2 (hCB2). There has also been research into its use in crop protection. Clifford et al. [47] reported on the insecticidal activity of alkamides found in the roots of *E. purpurea*. These alkamides exerted a mosquitocidal effect on the larvae of *Aedes aegypti,* such that at 100 and 10 µg/mL, 100% mortality was reported.

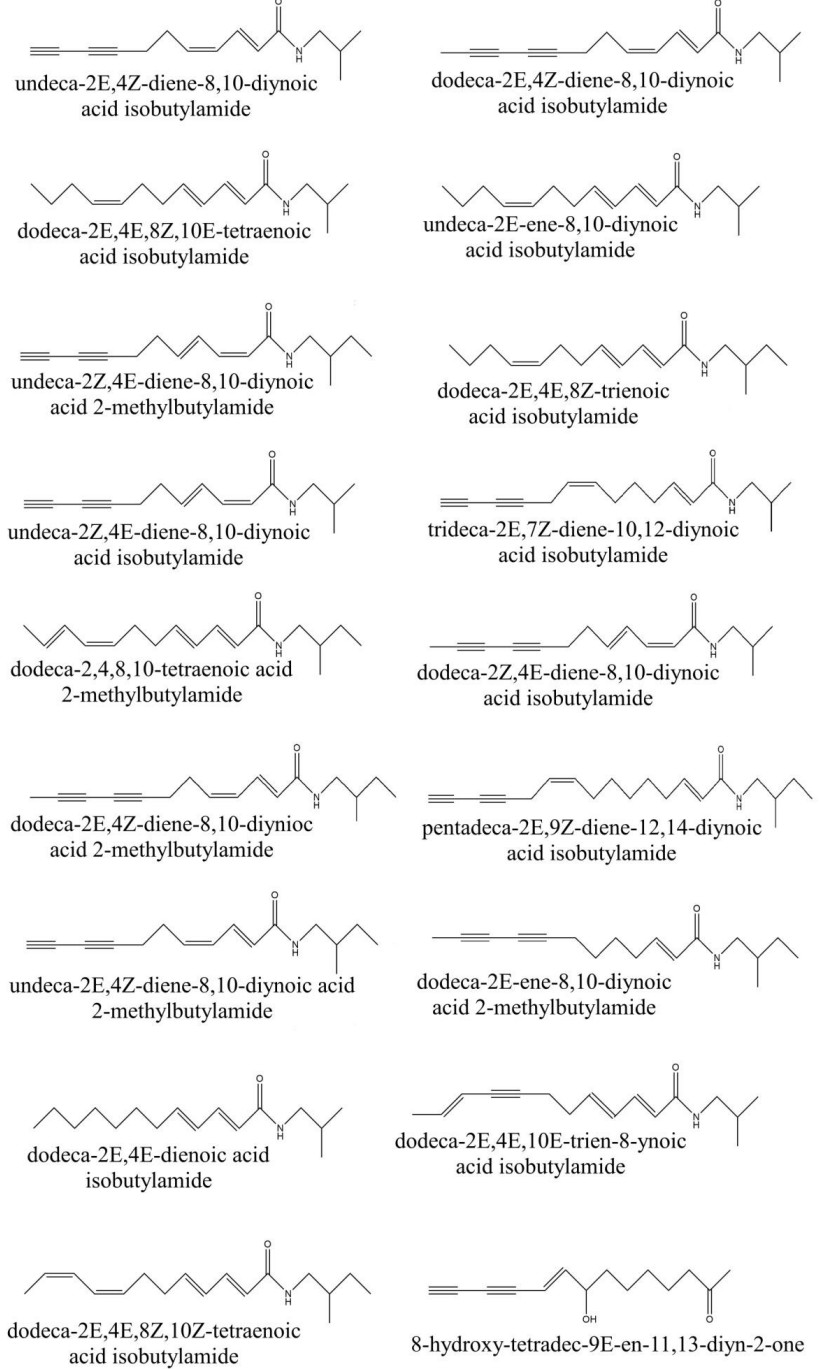

undeca-2E,4Z-diene-8,10-diynoic
acid isobutylamide

dodeca-2E,4Z-diene-8,10-diynoic
acid isobutylamide

dodeca-2E,4E,8Z,10E-tetraenoic
acid isobutylamide

undeca-2E-ene-8,10-diynoic
acid isobutylamide

undeca-2Z,4E-diene-8,10-diynoic
acid 2-methylbutylamide

dodeca-2E,4E,8Z-trienoic
acid isobutylamide

undeca-2Z,4E-diene-8,10-diynoic
acid isobutylamide

trideca-2E,7Z-diene-10,12-diynoic
acid isobutylamide

dodeca-2,4,8,10-tetraenoic acid
2-methylbutylamide

dodeca-2Z,4E-diene-8,10-diynoic
acid isobutylamide

dodeca-2E,4Z-diene-8,10-diynioc
acid 2-methylbutylamide

pentadeca-2E,9Z-diene-12,14-diynoic
acid isobutylamide

undeca-2E,4Z-diene-8,10-diynoic acid
2-methylbutylamide

dodeca-2E-ene-8,10-diynoic
acid 2-methylbutylamide

dodeca-2E,4E-dienoic acid
isobutylamide

dodeca-2E,4E,10E-trien-8-ynoic
acid isobutylamide

dodeca-2E,4E,8Z,10Z-tetraenoic
acid isobutylamide

8-hydroxy-tetradec-9E-en-11,13-diyn-2-one

**Figure 2.** Chemical structures of alkamides found in *Echinacea purpurea* (original).

## 2.2. Caffeic Acid Derivates

It was shown in the case of *E. purpurea* that cichoric acid was the main caffeic acid derivative (CAD) along with caftaric acid, while chlorogenic acid, echinacoside (found in the flowers, leaves and stem), and cynarin were present only in small amounts [31,48,49]. The CADs reported in *E. purpurea* are included in Figure 3.

In contrast to alkamides, which were found in rhizomes and roots of *E. purpurea*, CADs were more abundant in above-ground parts, such as flowers [1,50].

Phenolic compounds are thought to be responsible for antioxidant and antiviral activity [6,19,49,51–54]. Tsai et al. [55] reported that *Echinacea* extracts rich in cichoric acid induced apoptosis and reduced telomerase activity in intestinal cancer cells at concentrations of 200–500 µg/mL under in vitro conditions. In addition, Jeong et al. [56] have pointed out that cichoric acid in purple coneflower has antihyaluronidase activity and inhibits HIV-1 integrase and replication.

**Figure 3.** Chemical structures of caffeic acid derivatives (CADs) present in *Echinacea purpurea* (original).

## 2.3. Glycoproteins and Polysaccharides

Various polysaccharides have been described in *E. purpurea*. These components were first isolated by Wagner and Proksch [57], who detected two neutral fucogalactoxyloglucans (PS I, PS II) from the aerial parts, in which the main components of the fractions were arabinose, xylose, and galactose (PS I), and rhamnose, arabinose, xylose, and galactose (PS II), respectively [2]. Moreover, in a later research project [39,40] an acidic arabinogalactan, 4-O-methyl-glucuronoarabinoxylan, was identified from the hemicellulose fraction of *E. purpurea*. Another arabinogalactan protein was isolated from a suspension culture of *E. purpurea* containing a large amount of polysaccharide, the major monosaccharides being galactose and arabinose, a small protein fraction, and some glucuronic acid [41].

Polysaccharides and glycoproteins have been found in the aerial parts [58,59] of *E. purpurea*, such as in the leaves, stems and flowering tops, but also in the belove-ground parts [60–62], such as in the rhizomes and roots, but in general, research has tended to focus on the root parts.

Polysaccharides and glycoproteins are thought to be key components of the immunostimulatory properties of *E. purpurea* [8,31,63,64]. In addition to their immunostimulating effect, the antioxidant activity of polysaccharides has also been described [65].

Several studies have confirmed its positive effects under in vitro and in vivo immunological test systems. Wagner et al. [40] reported that fucogalactoxyglucan enhanced phagocytosis, as compared to arabinogalactan, which specifically stimulated macrophages to secrete tumor necrosis factor (TNF). The results of Burger et al. [66] supported this research and showed that the polysaccharide constituent of *Echinacea* increased the production of

interleukin-1 (IL-1) and interleukin-6 (IL-6) by macrophages in vitro. Luettig et al. [67] have shown that *E. purpurea*-derived polysaccharides stimulated T-cell activity more effectively than highly potent T-cell stimulators.

Barrett [68] has noted that polysaccharides from *E. purpurea* increased macrophage activity in mouse, rat and human experiments. Moreover, subsequent in vivo studies found that the purified polysaccharides provided protection against *Listeria monocytogenes* and *Candida albicans* infections by enhancing the activity of phagocytes [21,60,69,70].

Vimalanathan et al. [71] conducted a study that reported significant activity of polysaccharide-rich *Echinacea* extract against herpes simplex (HSV) and influenza virus (FV), which was also supported in a recent study by Liu et al. [72].

*2.4. Volatiles*

Volatile compounds can be detected in aerial parts and roots, with variable yields and chemical compositions [35,64,73,74]. The essential oil distilled from the roots of *Echinacea* species yields between 0.3 and 2.4% (*v/w*), although the aerial parts of the plants were observed to possess a smaller level of volatiles, yielding between 0.1 and 1.8% (*v/w*) [49,59,61,75].

According to Stănescu et al. [38], the volatile oil isolated from the aerial parts of *E. purpurea* contains mainly borneol, bornyl-acetate, caryophyllene, caryophyllene-epoxide, and germacrene D.

Schulthess et al. [35] conducted a thorough study of volatile compounds in ethanolic extracts of *E. purpurea* and essential oils from the achenes of three pharmaceutically approved species. The main components of *E. purpurea* were found to be α-pinene, β-farnesene, myrcene, limonene, carvomenthene, caryophyllene and germacrene D; however, it was highlighted that germacrene D, carvo-menthene and caryophyllene were characteristic of the achenes of *E. purpurea*.

Holla et al. [37] investigated the volatile oil content of *E. purpurea* flower heads grown in Slovakia. Seventy-two components were discovered, which consisted mainly of palmitic acid, germacrene D, α- and β-pinen.

Antimicrobial [76,77], antifungal [47,78], antioxidant [79] and antibacterial [80] properties of *E. purpurea* essential oil have been proven in numerous studies. According to Nyalambisa et al. [81], the essential oil from the roots of South African *E. purpurea* had strong anti-inflammatory and analgesic actions on rodents. Another study found that essential oil isolated from the flowers of *E. purpurea* had anti-inflammatory effects, as it greatly reduced the development of ear edema in rats [82].

Noorolahi et al. [83] found that *E. purpurea* essential oil extract had a strong antimicrobial effect on *Escherichia coli*, *Enterobacteriaceae*, and *Bacillus cereus*.

In a recent study, Teke and Mutlu [84] found that *Echinacea* oil, composed principally of β-cubebene and caryophyllene, caused 99.59% mortality of *Sitophilus granarius* 72 h after application, finally concluding that *Echinacea* oil has potential for use in the control of storage grain pests.

## 3. *Echinacea purpurea* Cultivation with Mutualistic–Symbiotic Microorganisms

In a current review, various *Echinacea* production biotechnologies were analyzed. Production of *Echinacea* in bioreactors and genetic engineering of plants can considerably increase, in a short time, the biomass and the content of active principles in plants, but these techniques are still too expensive, and large-scale production often decreases the yield. Polyploidy results in dwarf phenotypes in the case of *Echinacea* plants. The production of polyploid organisms is an unfeasible method due to the lower biomass of the plants, which results, on a larger scale, in lower yield of secondary metabolites and plant material [31].

The cultivation of entire plants appears to be the most profitable approach for the production of herbal products. In this context, the use of natural elicitors (e.g., growth regulators, natural stress response molecules) or mutualistic–symbiotic microorganisms

in semi-open field and open field cultures could be a cost-effective and environmentally friendly method to increase biomass and active principles in plants.

Most of the studies with *E. purpurea* that used elicitor induction were made in vitro (under controlled conditions), in bioreactors using the technology of wounded plants infected by *Agrobacterium rhizogenes*. The elicitors increased, besides biomass, the contents of cichoric acid, caftaric acid, alkylamides, anthocyanins, phenolics, flavanoids, and polysaccharides [6,31,85].

Although a large number of beneficial plants have been studied to observe the effects of mycorrhization, and despite the promising bioactive profile of *E. purpurea*, relatively few studies (a total of 5 studies) have been published with this species. The experiments about *E. purpurea* cultivation with mutualistic or symbiotic microorganisms are included in Table 1. In all cases, inoculation was made with *Rhizophagus irregularis* (except for one experiment when, besides *R. irregularis*, *Beauveria bassiana* and *Gigaspora margarita* were used) as single inocula or in combination with growth-promoting bacteria. Positive results were obtained both in greenhouse experiments [86–88] and in open-field cultivation [89,90]. Araim et al. [86] investigated the effect of AMF on the roots and shoots growth, but also on mineral, protein, alkamide, and phenolic acid contents of the concerned organs. Gualandi et al. [87,88] evaluated the pigment, sesquiterpene, and alkamide content of the leaves in addition to the plant growth characteristics.

In open-field conditions, results were reported on plant growth, nutrient content of the leaves and roots, and essential oil content of the roots [89,90].

The AMF and their associated microbiota increased not only plant biomass, but also the content of the following active principles: cichoric, caftaric, and chlorogenic acids, cynarin, alkilamides, essential oil and its compounds (beta-carophyllene, alpha-humulene, and germacrene-D).

**Table 1.** *Echinacea purpurea* cultivation with mutualistic microorganisms.

| Microorganisms | Experimental Setup | Effects | References |
|---|---|---|---|
| *Glomus intraradices* | - plants grown in a greenhouse, in pots filled with autoclaved sand/soil mixture (1:1, *v/v*) for 13 weeks | - increase in total mass, height, and leaves number<br>- increase in P and Cu content in the shoot<br>- increase in different phenolic acids content (cichoric, caftaric, and chlorogenic acids, and cynarin) in root<br>- increase in total phenolic acids content in shoot | [86] |
| *Glomus intraradices*, *Gigaspora margarita*, *Beauveria bassiana* | - plants grown in a greenhouse, in pots with calcined montmorillonite clay for 12 weeks | - increase in biomass and positive influence on plants development in severe nutrient deficiency stress, in the case of *G. intraradices*<br>- increase in cichoric and caftaric acids content in leaves in the case of *G. intraradices*, and in the whole plant (root + shoot) in the case of *G. intraradices*, and those treated with mycorrhiza in combination with *Beauveria bassiana*<br>- increase in relative concentration of two alkamides in roots in the case of plants treated only with *Beauveria bassiana*, high phosphorus and *B. bassiana*, and mycorrhyza in combination with *B. bassiana*<br>- increase in beta-carophyllene, alpha-humulene, and germacrene-D in leaves in the case of *G. intraradices*, high phosphorus and *B. bassiana*, and mycorrhyza in combination with *B. bassiana* | [87,88] |

**Table 1.** *Cont.*

| Microorganisms | Experimental Setup | Effects | References |
|---|---|---|---|
| *Azospirillum lipoferum*, *Azotobacter chroococcum*, *Pseudomonas fluorescens*, *Glomus intraradices* | - seedlings with 4–6 leaves transplanted and grown in open field conditions, on clay-loam soil, harvested after 6 months (October) in the first year, and in August in the second year | - combination of the three bacteria species with mycorrhiza resulted in significantly higher root and shoot mass, shoot length, larger number of branches, flower buds, and inflorescences, the results being similar with those obtained by NPK treatment<br>- treatments with mycorrhiza or with the mixture of the three bacteria increased the essential oil content of roots in the first year | [89] |
| *Rhizophagus irregularis*, *Pseudomonas fluorescens* | - 90-day-old seedlings transplanted in open field conditions, harvested after 6 months (November) in the first year, and in September in the second year | - increase in N, Cu, Fe and P contents in the case of mycorrhizal plants and those treated with *Pseudomonas fluorescens*<br>- increase in relative water content, plant height, leaf number, and leaf area index in the case of mycorrhizal plants<br>- increase in Zn content and plant height in the case of plants treated with *P. fluorescens*<br>- increase in the biological yield (g plant/m$^2$) in the case of mycorrhizal plants and those treated with *P. fluorescens* in the second year | [90] |

## 4. Plants from the Asteraceae Family Cultivated with Mutualistic–Symbiotic Microorganisms

Due to the limited number of published *Echinacea* studies with AMF species, this overview aims to exemplify similar research with plants from the Asteraceae family. One of the largest plant families is Asteraceae with more than 23,600 currently recognized species [91]. Plants from the Asteraceae family have several biochemical particularities, e.g., the presence of essential oils, glycosides, alkaloids, pyrethrins (sesquiterpene lactones), coumarins, latex with rubber resins, different bitter principles, inulin (polysaccharides) etc. [92].

A total of 13 articles were found in the selected period (2012–2022), reporting the results on growth and secondary metabolites of 17 plant taxa cultivated with AMFs.

### 4.1. Greenhouse Experiments

According to the collected articles, most of the experiments were conducted in greenhouse conditions, in plastic pots or root trainers. One of the research studies was conducted in a climatic chamber [93]. The substrates were sterilized using autoclave or steam, except for in Kheyri et al. [94] and Majewska et al. [95], who tried to partially simulate open-field conditions by using non-autoclaved soil. In the recently published articles, usually a larger number of AMF species were selected to investigate the effects of inoculations on different cultivars or micropropagated plants, in order to compare changes in growth and chemical compounds.

Kheyri et al. [94], in an experiment with *Calendula officinalis*, used as a single inocula, the following AMF species: *Glomus mosseae*, *G. intraradices*, *G. fasciculatum*, *G. caledonium*, *G. claroideum*, *G. versiform*, *G. geosporum*, *G. etanicatum*, and *G. gigaspora*. Plants were collected at 90 days after transplanting. The AMFs with the highest colonization rates were *G. mosseae*, *G. etanicatum*, and *G. geosporum*. These species had a significant positive effect on all measured parameters: plant growth and biomass, relative water content, photosynthesis pigments,

soluble sugars, soluble protein and antioxidant enzyme activities, antioxidant activity, mineral nutrient concentrations, total phenol content, and total flavonoid content. In another study with *Tagetes erecta* [96], application of *Rhizophagus irregularis*, *Claroideoglomus claroideum*, *Glomus hoi*, *Claroideoglomus etunicatum*, and *Acaulospora delicata* was made singly. Measurements were assessed at three months after inoculation. In comparison to the control, plant growth, metabolite production, and mineral nutrient concentrations all increased. Inoculation with *R. irregularis* produced the best results.

In several studies, micropropagated plants were cultivated to eliminate genotype differences, but also to obtain plants free of mycorrhiza. The secondary metabolite production depends greatly on the genotypes. Consequently, another tendency in studies was the comparison of several cultivars. In the case of two micropropagated cultivars of *Cynara cardunculus* var. *scolymus*, the effects of the following AMFs were compared: *Funneliformis mosseae*, *Rhizoglomus irregulare*, *Claroideoglomus claroideum*, and an undescribed *Glomus* sp. [97]. Plants were grown from March until September. The measured parameters were total phenolic compounds, antioxidant activity, chlorophyll content, and mineral content. Among the studied AMFs, *Claroideoglomus claroideum* significantly increased the total phenolic content and the chlorogenic acid content of both cultivars. Antioxidant activity was enhanced by *C. claroideum* and *Funneliformis mosseae*. In another study, *Mikania glomerata* and *M. laevigata* plants propagated vegetatively using stem cuttings were cultivated for 8 weeks [98], after inoculation with *Rhizophagus irregularis*. Besides the positive effects of the fungi (significant increase in foliar biomass and diterpene kaurenoic acid content in *M. laevigata*, mineral content in leaves of both species), it was observed that AMFs significantly reduced the tricaffeoylquinic acid contents in leaves of *M. glomerata*.

Avio et al. [99] investigated the effect of *Rhizophagus irregularis* and *Funneliformis mosseae* on two cultivars of *Lactuca sativa var. crispa*. Changes in total phenolics and antioxidant activities differed by AMFs and cultivars. *R. irregularis* significantly increased total phenolic content and antioxidant activity of both cultivars, while anthocyanins content was enhanced by both AMF species only in the red leaf cultivars.

In addition, experiments with a mixture of AMF species were conducted. Although there were more flowers in the case of *Calendula officinalis 'Calypso Orange with Black Eye' cultivar* due to the AMF colonization (mixture of *Claroideoglomus etunicatum*, *C. claroideum*, and *Rhizophagus irregularis*), these flowers were smaller. Hence, the flower production was not significantly different from control plants. The AMF mixture increased the concentration of two of the eight phenolic compounds identified in the flowers of Marigold plants, but the major phenolic constituents were not increased [100].

The effects of *Rhizophagus irregularis*, *Funneliformis mosseae*, and *Claroideoglomus claroideum* in different soil types were evaluated in the case of two invasive species: *Rudbeckia laciniata* and *Solidago gigantea* [95]. Soils were collected from two characteristic habitats. The growth of the plants, concentration of phosphorus, and photosynthetic performance were measured after 3 months. *C. claroideum* showed no effect on *R. laciniata*. Both *R. irregularis* and *F. mosseae* increased the biomass of *Rudbeckia* plants. However, the results varied according to the type of soil. Although the variations were more apparent in the fallow soil, *Solidago gigantea* responded favorably to all applied AMF species, with *R. irregularis* being the most successful in enhancing biomass. Photosynthetic performance of both plants was influenced only by the soil type, while P concentration in shoots and roots was dependent on AMFs and soil types. The best results in P enhancement were obtained with *R. irregularis*.

AMFs were used to enhance growth and flower quality of two ornamental species: *Chrysanthemum morifolium* and *Tagetes erecta* [101]. The experiment was conducted in a shade net, on pots filled with sterile soil. Inoculation was made with pure cultures of *Acaulospora laevis*, *A. scrobiculata*, *Glomus coremioides*, *G. intraradices G. fasciculatum*, *G. mannihotis*, and *Gigaspora albida*. All AMF treatments significantly enhanced the early flowering and the number of flowers in the case of *C. morifolium*. Dry weights of both plant species were significantly higher due to the treatments (except for *T. erecta* plants treated with *A. laevis*). *G. intraradices* proved to be more efficient for the increase in flower number in the case

of *T. erecta*. Root length, plant height and moisture-retaining ability of flowers were also enhanced by the different AMF species depending of the host plants.

Not all research studies reported positive results. Duc et al. [93] in an experiment with salinity stress, besides others, examined how several single AMF species—including *Funneliformis mosseae*, *Septoglomus deserticola*, and *Acaulospora lacunosa*—as well as a combination of six AMF species (*Claroideoglomus etunicatum*, *C. claroideum*, *Rhizoglomus microaggregatum*, *Rhizophagus intraradices*, *Funneliformis mosseae*, and *F. geosporum*) affected the growth and the physio-biochemical traits of the *E. prostrata* plant under non-saline environments. *S. deserticola* significantly decreased fresh shoot weight after 4 and 8 weeks of plant growth. At 4 weeks, *F. mosseae*, *S. deserticola*, and *A. lacunosa* significantly decreased plant height in comparison to the control plants. After 8 weeks, *S. deserticola* significantly reduced the total phenolic concentration in comparison to control plants and other single AM treatments. When compared to control plants, AMFs did not increase proline content under non-stressful situations.

### 4.2. Semi-Open Field and Open Field Experiments

Only a few studies were published with AMF application in semi-open field (plants grown outside in pots) and open field conditions. Besides the frequently used species (*Rhizophagus irregularis* and *Funneliformis mosseae*), inoculations with other AMFs (*Septoglomus viscosum*, *Acaulospora laevis*, *A. scrobiculata*, *Glomus coremioides*, *G. fasciculatum*, *G. mannihotis*, and *Gigaspora albida*) and AMF mixtures were also attempted.

In an experiment with *Stevia rebaudiana* [102], three micropropagated chemotypes were cultivated on loamy soil (in southern Italy) for three vegetation periods. The measured parameters were: diterpene glycosides content, dry leaf and stem yield. *Septoglomus viscosum* only improved leaf yield, especially in the case of two chemotypes.

*Artemisia annua* plants grown outside (in Romania), in pots filled with sterile peat, inoculated with *R. irregularis*, showed an increase of 30% in fresh and dry biomass compared to control plants [103]. Compared to untreated plants, those colonized with AMF had significantly higher artemisinin content, with an average of 17%. The glandular hairs were shown to be significantly correlated with artemisinin concentration. Similar experiments were conducted in pots with three characteristic soil types from the region (endostagnic argic Luvisol, stagnic colluvic Gleysol, and stagnic gleyic Anthrosol) and sterile peat [104]. Plants were grown outside. *R. irregularis* significantly improved the artemisinin and essential oil content of the potted plants grown in all of the studied substrates. Biomass was increased considerably by AMFs only on sterile peat. In the same experimental year, inoculated plants were also grown in open field, on stagnic vertic Luvisol. Inoculation caused a significant decrease in stem biomass compared to the control. While no differences were observable in the case of artemisinin content, the essential oil content was increased. When comparing the essential oil yield of AMF-treated plants grown in pots to those grown in open field soil, no significant differences were detected. Differences in chemical compound abundance were registered (higher concentrations of beta-farnesene and germacrene D in AMF plants).

Bączek et al. [105] used an AMF mixture from *Rhizophagus intraradices* (BEG140), *Funneliformis mosseae*, *Claroideoglomus claroideum* (BEG 210), *Funneliformis geosporum*, *Rhizophagus intraradices* (SAMP7), and *Claroideoglomus claroideum* (E10) for the cultivation of *Matricaria recutita* on alluvial soil (in Poland). Plant growth parameters (fresh mass of the root and herb, number of flowering shoots) and phenolic compound content of pharmaceutical importance (chlorogenic, caffeic, ferulic, and rosmarinic acids as well as apigenin-7-O-glucoside, isorhamnetin, and luteolin-4′-glucoside) were enhanced in the inoculated plants. The essential oil yield and composition were not affected by the inoculation.

Based on the above studies with *E. purpurea* relatives and their interactions with AMFs, several research directions have been outlined in order to improve the growth and production of secondary metabolites in *E. purpurea*. Considering the biodiversity of the AMF species, comparative experiments are recommended for evaluation of their effects

on the host plant. In addition, inoculation with AMF mixtures should be attempted to assess the influence of synergistic activities on plant development. It would be important to test various *E. purpurea* cultivars because variations in genotype have a significant impact on secondary metabolites. The positive effects of mutualistic–symbiotic microorganisms on host plants were described in the case of several medicinal and crop plants. The most utilized AMF fungi were *Rhizophagus irregularis* and *Funneliformis mosseae* [22]. Since the effect of AMF depends on the variety of field conditions, more research on the application of different AMF (singly and in various combinations with mycorrhiza helper/plant growth promoter bacteria) to *Echinacea* plants growing in the field would be necessary.

## 5. Mycorrhiza's Mechanism of Action in Secondary Metabolite Production

In the case of plant elicitors, the proposed mechanisms for increase in CADs, phenolics, and flavonoids is through the phenylpropanoid pathway, activated by the defense response of the plants and up-regulation of the genes involved in these processes [31,106].

Kapoor et al. [107] and Kumar et al. [30] in their comprehensive reviews have synthesized the possible mechanisms of AMF in enhancing the production of chemical compounds in host plants. The following interconnected mechanisms were proposed: biomass increase in different organs (root, shoot, leaves etc.), enhancement of phosphorus, nitrogen and micronutrients (Mn, Mg, Fe, Cu, Zn) uptake, alteration of the phytohormones synthesis, increase in the glandular hairs number, activation of different biosynthesis pathways, and changes in gene expressions.

The mycorrhizal relationship can result in changes in the concentration of jasmonic acid (JA), gibberellic acid ($GA_3$), abscisic acid (ABA), and cytokinins (CKs). These plant hormones play an important role in the formation and development of glandular hairs [107]. According to Maes et al. [108], JA increased the size and number of glandular hairs, stimulated the biosynthesis of artemisinin precursors (artemisinic acid, dihydroartemisinic acid), and elevated the expression of the genes responsible for the biosynthesis of artemisinin. It was also shown that the production of JA in *Artemisia annua* increased due to the inoculation with *Rhizophagus irregularis* [109].

Schweiger and Müller [110] suggest that plants' metabolic responses can be nutrient-mediated (higher P or N uptake from nutrient-rich environment with or without symbiosis) or from direct AMF effects (independent of nutrient supply) relying on symbiosis.

Molecular understanding of the mechanism of action of AMF and associated bacteria has been the subject of recent studies. Several experiments have been carried out to investigate the effects of mycorrhiza on gene expression in different plant species of global importance [30,107,111–113]. According to these results, secondary metabolite production and growth of the economic plants can be affected by mutualistic, symbiotic or parasitic microorganisms via upregulation of the genes involved in phytohormonal synthesis, glandular hair formation, and in different biosynthesis pathways, e.g., the mevalonate (MVA), methyl erythritol phosphate (MEP), and phenylpropanoid pathways (Table 2).

The synthesis of steviol-glycosides in *Stevia rebaudiana* starts via the MEP pathway, such as the synthesis of artemisinin and different terpenes in *A. annua* [114]. In the case of *A. annua*, the two precursors of farnesyl diphosphate (FDP)—isopentenyl diphosphate (IDP) and dimethylallyl diphosphate (DMADP)—are synthesized in plastids via the MEP pathway and in the cytosol via the MAV pathway [115]. AMFs act on the bioactive metabolic products of *A. annua* through the MEP pathway (and not through the MVA pathway), while JA has an important role in the mechanism of artemisinin synthesis [109]. The transcript level of MEP pathway genes significantly increased in AMF-treated *S. rebaudiana* plants compared to the control during the synthesis of steviol-glycosides [109,116]. Both experiments were conducted in semi-open field conditions, in pots filled with autoclaved sandy loam soil.

Adolfsson et al. [111] investigated gene expression in *Medicago truncatula* mycorrhized with *Rhizophagus irregularis*, where the results were compared with those obtained for phosphorus-treated (non-AMF-inoculated plants) and control plants. The experiment was

conducted in a growth chamber. Plants were cultivated in pots with Agsorb substrate. The expressions of genes involved in phenylpropanoid, flavonoid, terpenoid, lipid, JA, ABA, CK biosynthesis, and the *MYC2* gene (the main regulator of JA-dependent responses) were increased during mycorrhizal treatment. CK levels were increased in both treatments (mycorrhizal and phosphorus), while ABA levels were increased only in mycorrhizal plants. In addition, it was reported that control plants foliar-treated with either ABA or JA induced *MYC2* expression, while the expressions of flavonoid and terpenoid biosynthetic genes were induced only by JA. Shoot development was explained by CK action. ABA signaling pathways had a probable role in the defense strategies under biotic and abiotic stresses.

Xie et al. [112] inoculated *Glycyrrhiza uralensis* with *Rhizophagus irregularis* and hypothesized that the treatment would increase the amount of the two important bioactive compounds, glycyrrhizin and liquiritin, but also the expression of genes involved in the synthesis of the compounds. In this experiment, plants were cultivated in a growth chamber, in pots containing autoclaved sand and soil (1:2, *v:v*). The synthesis of glycyrrhizin is manly initiated by the MVA pathway. AMF significantly increased the expression of all studied genes involved in glycyrrhizin and liquiritin biosynthesis compared to controls, when an adequate amount of water was administrated to the plants. A significant increase in gene expression was also obtained when plants were exposed to moderate drought stress, except for the *HMGR* gene.

**Table 2.** Studies reporting mutualistic, symbiotic or parasitic microorganisms that affect gene transcriptions in plants.

| Host Plant | Inocula | Upregulated Genes | Role of the Genes | References |
|---|---|---|---|---|
| *Artemisia annua* | *Rhizophagus irregularis* | *TTG1* | synthesis of a transcription factor involved in the formation of glandular hairs | [114] |
| | | *DXS1* | formation of 1-deoxy-D-xylulose-5-phosphate (DXP) in the MEP pathway | |
| | | *DXR* | formation of 2-C-methyl-D-erythritol-4-phosphate (MEP) | |
| | | *ADS*, *CYP71AV1*, *DBR2*, *ALDH1* | biosynthesis of artemisinin | |
| *Stevia rebaudiana* | *Rhizophagus irregularis* | *MDS* | synthesis of 2-C-methyl-D-erythritol-2,4-cyclodiphosphate synthase (MDS), a key enzyme in the MEP pathway | [109] |
| | | | the first stage of the synthesis of steviol glycosides: | |
| | | *DXS1* | formation of 1-deoxy-D-xylulose-5-phosphate (DXP) in the MEP pathway | |
| | | *DXR* | formation of 2-C-methyl-D-erythritol-4-phosphate (MEP) | |
| | | | the second stage of the synthesis of steviol-glycosides | |

**Table 2.** *Cont.*

| Host Plant | Inocula | Upregulated Genes | Role of the Genes | References |
|---|---|---|---|---|
| | | *GGDPS* | synthesis of geranylgeranyl diphosphate synthase (GGDPS) | |
| | | *CPPS, KS, KO, KAH* | formation of steviol | |
| | | | the third stage of the synthesis of steviol glycosides: | |
| | | *UGT85C2, UGT74G1, UGT76G1* | glycosylation of steviol | |
| *Stevia rebaudiana* | *Piriformospora indica* | *DXR, GGDPS, KS, KO UGT85C2, UGT74G1, UGT76G1* | same roles as above | [116] |
| *Medicago truncatula* | *Rhizophagus irregularis* | *3′GT, DFR, CHMT, 4CL, LDOX, ANTHOCYANIN 5-AROMATIC ACYLTRANS-FERASE, ANTHOCYANIDIN 3-O-GLUCOSYL-TRANSFERASE, ISOFLAVONOID GLYCOSYL-TRANSFERASE, ISOFLAVONOID MALONYL TRANSFERASE, CAFFEATE 3-O-METHYL-TRANSFERASE* | synthesis of phenylpropanoids, flavonoids and anthocyanins | [111] |
| | | *TERPENE SYNTHASE1, HMG-CoA REDUCTASE, SQE3,UGT73K1, CYP76A61, CYP93E2, CYP72a67v2* | terpenoid biosynthesis | |
| | | *TRIACYLGLYCEROL LIPASE, DGAT* | lipid biosynthesis | |
| | | *9-LOX, 13-LOX, AOS, AOC,* MYC2, *JAZ* | jasmonic acid (JA) biosynthesis | |
| | | *HOMEOBOX-LEU ZIPPER ATHB-7, ZEAXANTHIN EPOXIDASE* | abscisic acid (ABA) biosynthesis | |
| | | *CYTOKININ-O-GLUCOSYL-TRANSFERASE* | cytokinin (CK) biosynthesis | |

**Table 2.** *Cont.*

| Host Plant | Inocula | Upregulated Genes | Role of the Genes | References |
|---|---|---|---|---|
| *Glycyrrhiza uralensis* | *Rhizophagus irregularis* | HMGR | synthesis of mevalonic acid (MVA) in the mevalonate pathway | [112] |
| | | SQS1, β-AS, LUP | formation of squalene, β-amyrin and lupeol | |
| | | CYP88D6, CYP72A154 | formation of glycyrrhetinic acid, the precursor of glycyrrhizin | |
| | | CHS | formation of liquiricin | |
| *Helianthus annuus* | *Rhizophagus irregularis* and *Rhizoctonia solani* parasitic fungi | PAL1, C4H | phenylpropanoid synthesis | [113] |
| | | CHS, CHI2, F3H, FLS1, DFR, F30H | synthesis of flavonoids | |
| | | AN1, AN2 | conversion of anthocyanidin to anthocyanin | |
| | | HCT, HQT, C3H | synthesis of chlorogenic acid | |

Rashad et al. [113] investigated the effect of a symbiotic fungus (*Rhizophagus irregularis*) and a parasitic fungus (*Rhizoctonia solani*) on the temporal changes in the expression of major genes involved in phenylpropanoid, flavonoid and chlorogenic acid biosynthetic pathways in *Helianthus annuus*. The experiment was conducted in a greenhouse, in pots filled with clay–sand soil (1:2). In total, the expressions of 13 genes (known to play important roles in phenylpropanoid, flavonoid and chlorogenic acid synthesis) were studied. In their experiment, three types of treatments were used in addition to the control: plants infected with *R. solani*, plants colonized with *R. irregularis* and plants treated with both fungal species. Their results showed that all three treatments induced the transcriptional expression of the targeted genes, but the double treatment showed a better induction effect compared to the single treatments.

## 6. Conclusions

Studies have shown in several plant species that the mutualistic–symbiotic relationship between AMF and/or growth-promoting bacteria and host plants can increase the accumulation of primary and secondary metabolites and improve plant morphological parameters. In the case of *E. purpurea*, only few studies were published on this topic. The AMF inocula used in the experiments were mainly *Rhizophagus irregularis*. The studies found in the selected period (2012–2022) reported the effects of 21 AMFs used as single inocula or as a mixture on growth and secondary metabolites of 17 plant taxa from the Asteraceae family.

However, these studies, with *E. purpurea* inclusive, have mostly been carried out under controlled conditions, in greenhouses, or in vitro in sterile environments, but it should be taken into consideration that the efficiency of the mutualistic–symbiotic relationship can be greatly influenced by environmental factors and the soil microbiome of the cultivation area.

In this context, future research to explore the potential of mycorrhizal and associated microorganisms in the cultivation of *E. purpurea* or other economic plants could be directed toward its application in field crop production. In order to use different AMF species and other microorganisms as a biofertilizer for targeted inoculation of growing substrates in

agriculture, experiments are needed to assess the ability of selected species to compete with native microorganisms.

For the identification of the most effective synergistic combinations of AMF and related bacterial populations, transcriptomic and metabolomic investigations might also be useful.

**Author Contributions:** Conceptualization, M.I.; investigation, M.I. and E.D.; writing—original draft preparation, M.I.; writing—review and editing, K.B., K.M., E.K., E.B. and F.V.D.; supervision, F.V.D.; funding acquisition, F.V.D. All authors have read and agreed to the published version of the manuscript.

**Funding:** This work was supported by a grant from the Ministry of Research, Innovation and Digitization, CNCS-UEFISCDI, project number PN-III-P4-PCE-2021-0750, within PNCDI III.

**Data Availability Statement:** Not applicable.

**Conflicts of Interest:** The authors declare no conflict of interest.

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
