# Peer review of "The Importance of Mycorrhizal Fungi in the Development and Secondary Metabolite Production of Echinacea purpurea and Relatives (Asteraceae): Current Research Status and Perspectives"

_horticulturae, doi:10.3390/horticulturae8121106_

Round 1
Reviewer 1 Report
After carefully reading the manuscript I have only a few suggestions.
Please explain AMF when first mentioning it. Not every reader is acquainted with the term.
In order to collect the recent findings about how the inoculation with AMF affects gene expressions in different host plants, all time literature was used. Please be specific here, is it recent or is it all-time.
Line 204: correct In a currently review to In a current review
Reviewer 2 Report
The present manuscript is an interesting review.
Some minors aspects to be considered before accepting for publication:
When referring to the cultivation of E. purpurea, in order to increase biomass and economic importance secondary metabolite production in plants, please elaborate a bit more about.
Please add a short paragraph describing the chemical compounds reported in Echinacea purpurea before describing them in details.
Lines 97-98: "The specialty literature included in this overview was collected from the Science Direct, Springer Link and Google Academic platform." - how many articles?
Lines 98-106: please specify the nummber of articles curated/found/selected
I suggest to draw a schema explaining the steps following the literature research and articles colletion, related to lines: 100-107 to be more clear the process.
Line 204: please add reference
Lines 208-209: "Thus currently is an unfeasible method due to the decreased biomass production" - not clear, please reformulate
Lines 247-257: based on the statement: "Due to the limited number of published Echinacea studies with AMF species, the overview aims to exemplify similar researches with plants from the Asteraceae family." - i would suggest to modify the title accrodingly, as actually a pretty small part of this review focuses on Echinacea studies. Actually this issue has been presented in the abstract as well.
Round 2
Reviewer 2 Report
The authors adressed my suggestions/comments. Thank you.